# The Association between Prognostic Nutritional Index (PNI) and Intraoperative Transfusion in Patients Undergoing Hepatectomy for Hepatocellular Carcinoma: A Retrospective Cohort Study

**DOI:** 10.3390/cancers13112508

**Published:** 2021-05-21

**Authors:** Ji Hoon Sim, Sung-Hoon Kim, In-Gu Jun, Sa-Jin Kang, Bomi Kim, Seonok Kim, Jun-Gol Song

**Affiliations:** 1Department of Anesthesiology and Pain Medicine, Asan Medical Center, University of Ulsan College of Medicine, Seoul 05505, Korea; atlassjh@hanmail.net (J.H.S.); shkimans@gmail.com (S.-H.K.); igjun@amc.seoul.kr (I.-G.J.); sajinkg@naver.com (S.-J.K.); dudnd5@gmail.com (B.K.); 2Department of Clinical Epidemiology and Biostatistics, Asan Medical Center, University of Ulsan College of Medicine, Seoul 05505, Korea; seonok@amc.seoul.kr

**Keywords:** prognostic nutritional index, bleeding, transfusion, hepatectomy, hepatocellular carcinoma

## Abstract

**Simple Summary:**

The prognostic nutritional index (PNI), which describes a patient’s nutritional, inflammatory status, and immune response, has been reported as a predictor associated with prognosis in a variety of cancers, and has been reported to be associated with surgical outcomes in patients with hepatocellular carcinoma (HCC). However, few studies have assessed the association between PNI and intraoperative transfusion. This study evaluated the predicting value of preoperative PNI for intraoperative transfusion in patients who underwent hepatectomy for HCC. We found that preoperative PNI < 44 was significantly associated with intraoperative transfusion and surgical outcomes. These results suggest that preoperative PNI might be a predictor of intraoperative transfusion, and surgical prognosis in patients who underwent hepatectomy.

**Abstract:**

Background: PNI is significantly associated with surgical outcomes; however, the association between PNI and intraoperative transfusions is unknown. Methods: This study retrospectively analyzed 1065 patients who underwent hepatectomy. We divided patients into two groups according to the PNI (<44 and >44) and compared their transfusion rates and surgical outcomes. We performed multivariate logistic and Cox regression analysis to determine risk factors for transfusion and the 5-year survival. Additionally, we found the net reclassification index (NRI) to validate the discriminatory power of PNI. Results: The PNI <44 group had higher transfusion rates (adjusted odds ratio [OR]: 2.20, 95%CI: 1.06–4.60, *p* = 0.035) and poor surgical outcomes, such as post hepatectomy liver failure (adjusted [OR]: 3.02, 95%CI: 1.87–4.87, *p* < 0.001), and low 5-year survival (adjusted OR: 1.68, 95%CI: 1.17–2.24, *p* < 0.001). On multivariate analysis, PNI <44, age, hemoglobin, operation time, synthetic colloid use, and laparoscopic surgery were risk factors for intraoperative transfusion. On Cox regression analysis, PNI <44, MELD score, TNM staging, synthetic colloid use, and transfusion were associated with poorer 5-year survival. NRI analysis showed significant improvement in the predictive power of PNI for transfusion (*p* = 0.002) and 5-year survival (*p* = 0.004). Conclusions: Preoperative PNI <44 was significantly associated with higher transfusion rates and surgical outcomes.

## 1. Introduction

Hepatocellular carcinoma (HCC) is one of the most common types of primary cancer, the sixth most common neoplasm, and the third leading cause of cancer death [1]. The incidence of HCC is increasing due to hepatitis C virus (HCV) infection, with more than 782,000 new cases diagnosed each year, and recently, non-alcoholic fatty liver disease (NAFLD) has become an important cause of hepatocellular carcinoma in developed areas. [1,2]. The main treatment is hepatectomy, which is one of the most complex major abdominal surgeries [3]. Although various surgical techniques, such as laparoscopy, the pringle maneuvers, and inferior vena cava (IVC) clamping to minimize bleeding during liver surgery, have reduced intraoperative bleeding [3,4], hepatectomy still has a very high bleeding risk and often requires blood transfusion [5,6,7].

The prognostic nutritional index (PNI), which is calculated from serum albumin concentration and total lymphocyte count, describes a patient’s nutritional and inflammatory status, and immune response [8]. PNI has been reported as a prognostic predictor in a variety of cancers [9,10], and a recent meta-analysis reported that preoperative PNI is a prognostic marker in HCC patients [11]. Further, PNI has been reported to be significantly associated with survival and complications in surgical patients [12,13]. However, there are few studies on the association between preoperative PNI and intraoperative bleeding and transfusion in surgical patients.

Therefore, we sought to validate preoperative PNI as a predictor for intraoperative transfusion in patients who underwent hepatectomy for hepatocellular carcinoma.

## 2. Materials and Methods

### 2.1. Study Design and Patient Population

The institutional review board (IRB) of Asan Medical Center (protocol number: 2021-0243) approved this study and the need for written informed consent was waived due to the retrospective nature of our study. We reviewed all patients diagnosed with primary HCC based on the International Classification of Diseases, tenth revision (ICD-10) who underwent open or laparoscopic hepatectomy between January 2008 and October 2015. Adult patients aged 18 years and older were included in the study. All surgeries were performed sequentially by one surgeon. The exclusion criteria were as follows: age < 18 or ≥80 years; patients who had hematologic disease; patients who had taken anticoagulants, such as warfarin and aspirin; patients who underwent emergency surgery; pregnant women; and patients with incomplete data or missing PNI values.

### 2.2. General Anesthesia

After routine monitoring (pulse oximetry, electrocardiography, capnography, and noninvasive blood pressure measurement), general anesthesia was induced with an intravenous bolus injection of thiopental sodium (4–5 mg/kg) and rocuronium (0.6–1.2 mg/kg). Before endotracheal intubation, an intravenous bolus of fentanyl, 1– 2 μg/kg, was injected and anesthesia was maintained using sevoflurane 2–4 vol% in 50% air/oxygen. The patients were mechanically ventilated with a tidal volume of 6–8 mL/kg, and the respiratory rate was adjusted to maintain an end-tidal carbon dioxide partial pressure of 35–40 mmHg. Invasive arterial and central venous pressure monitoring were routinely conducted. During anesthesia, crystalloid solutions (lactate Ringer’s solution or plasma solution) or colloid solutions (5% albumin, synthetic colloid (Voluven^®^; Fresenius Kabi, Bad Homburg, Germany) were administered. The total volume of infused synthetic colloid did not exceed 20 mL/kg. If the plasma hemoglobin (Hb) level was less than 8 g/dL during surgery, packed red blood cells (RBCs) were transfused. Vasopressors and inotropes, such as ephedrine, phenylephrine, or norepinephrine, were administered if the mean arterial blood pressure was <65 mmHg, as determined by an anesthesiologist.

### 2.3. Surgical Technique

The patient selected the surgical method after receiving a comprehensive explanation of laparoscopic surgery and open surgery. For laparoscopic surgery, the patient was placed in the supine position and the reverse Trendelenburg position. The first access to the abdomen was through an umbilical port. After a pneumoperitoneum by carbon dioxide was established with a pressure of 10 to 15 mmHg, 3 or 4 additional ports were placed. For open surgery, abdominal surgery was performed through a J-shaped incision. The liver parenchyma was separated using an ultrasonic surgical aspirator (CUSA Excel; Valleylab, Boulder, CO, USA). After parenchymal incision, the hepatic vein was cut with a vascular stapler or clamp, and then sutured. After parenchymal dissection, the hepatic vein was ligated with an endoscopic stapler. The resected specimen was retrieved through a Pfannenstiel incision.

### 2.4. Clinical Data Collection and Outcome Assessments

Demographic data, preoperative laboratory, and intraoperative variables were collected by using our computerized medical record system. Demographic data included age, sex, weight, body mass index (BMI), liver cirrhosis, tumor etiology, antiviral therapy, preoperative transarterial chemoembolization (TACE), TNM staging, diabetes mellitus (DM), hypertension (HTN), coronary artery disease (CAD), model for end-stage liver disease (MELD) score, and Child–Turcotte–Pugh (CTP) score. Variables related to the patients’ tumor characteristics included the number of tumors, largest tumor size, lymph node invasion, and metastasis.

Preoperative laboratory values included preoperative white blood cell (WBC), Hb, platelet, prothrombin time, serum creatinine (sCr), estimated glomerular filter ratio (eGFR), albumin, total bilirubin, aspartate aminotransferase (AST), alanine aminotransferase (ALT), and sodium. Serum creatinine levels were checked daily from postoperative day 1 (POD) to 7 to confirm acute kidney injury (AKI). The preoperative PNI, and red cell distribution width (RDW) values were also collected. PNI was calculated using the following formula: {10 × serum albumin (g/dL)} + {0.005 × total lymphocyte count (per mm^3^)}. Intraoperative variables included operation time, laparoscopic surgery, extensive surgery (≥3 segments), administered fluids, and urine output. Intraoperative RBC, fresh frozen plasma (FFP), platelet transfusion, postoperative transfusion, hospital day, postoperative AKI, post hepatectomy liver failure (PHLF), intensive care unit (ICU) admission, prolonged ICU stay (≥2 days), and 1-year and 5-year survival records were also collected.

### 2.5. Primary and Secondary Outcomes

The primary outcomes were to compare the incidence of intraoperative transfusion according to the preoperative PNI (<44 and >44) and to analyze the risk factors associated with intraoperative transfusion. Massive transfusion was defined as ≥10 units of packed RBC (pRBC) within 24 h or ≥4 units of pRBC within 1 h [14]. The secondary outcomes were to compare the surgical outcomes, such as hospital day, postoperative AKI, PHLF, ICU admission, prolonged ICU stay (≥2 days), and 1-year and 5-year survival, according to the preoperative PNI cut-off value. AKI was defined by Kidney Disease Improving Global Outcomes classification (KDIGO): sCr increased by at least 1.5 times at baseline before surgery within 7 days or sCr increased by 0.3 mg/dL within 48 h [15]. Additionally, we evaluated the predictive power of PNI for intraoperative transfusion through net reclassification index (NRI) analysis.

### 2.6. Statistical Analysis

Categorical variables were analyzed using chi-square or Fisher exact test, and the independent t test or Mann–Whitney U test was used to evaluate continuous variables, as appropriate. Continuous variables are presented as mean ± standard deviation (SD), median with interquartile range, or n (proportion). We performed a receiver operating characteristic (ROC) curve analysis to determine the cut-off value for predicting intraoperative transfusion. To compare the PNI discrimination power for predicting intraoperative transfusion, a clinical model was created with the previously known risk factors (age, MELD scores, and Hb) and PNI was added to calculate the area under the curve (AUC), 95% confidence intervals (CI), and net reclassification index (NRI). The NRI is a measure for evaluating the improvement in prediction performance gained by adding a marker to a set of baseline predictors [16]. NRI was used to comprehensively evaluate the discrimination of the model along with the AUC [16].

We used multiple logistic regression analysis to determine independent predictors of intraoperative transfusion and all variables with *p*-value less than 0.1 in univariate analysis were included in the multivariate analysis. To assess the hazards ratio (HR) of the risk factors of 5-year survival, Cox regression analysis was also used. The Kaplan–Meier (KM) method was used to analyze the cumulative survival rate between two groups (PNI level <44 group and ≥44 group), and the log rank test was used to evaluate the change between curves. All *p*-values less than 0.05 were considered statistically significant. All statistical analyzes were performed using IBM SPSS Statistics version 22.0 for Windows (IBM Corp., Armonk, NY, USA).

## 3. Results

Out of 1167 patients who underwent hepatectomy, 102 patients were excluded (patients who had hematologic disease (*n* = 4); patients who had taken anticoagulants, such as warfarin and aspirin (*n* = 42); patients who underwent emergency surgery (*n* = 12); and patients with incomplete data or missing PNI values (*n* = 44), 1065 patients were included in this study. According to ROC curve analysis, a cut-off value of preoperative PNI of 44 predicted intraoperative transfusion (area under the curve 0.733, sensitivity 68.3%, specificity 71.0%) (Figure 1). Patients were divided into two groups by PNI value: PNI <44 (*n* = 333) and PNI ≥44 (*n* = 732).

The demographic, laboratory, and intraoperative variables of the patients are shown in Table 1. The preoperative PNI <44 group were older (*p* = 0.026), more likely to be female (*p* = 0.002), had a lower BMI (*p* = 0.038), poor TNM staging (*p* = 0.046), and higher MELD scores (*p* < 0.001) and CTP scores (*p* < 0.001) than the preoperative PNI group ≥44. However, the incidence of liver cirrhosis (*p* = 0.764), DM (*p* = 0.487), HTN (*p* = 0.163), CAD (*p* = 0.728) and tumor etiology (*p* = 0.623), antiviral therapy (*p* = 0.866), and preoperative TACE (*p* = 0.542) were not significantly different between the two groups.

Regarding the laboratory variables, the PNI <44 group had significantly lower levels of WBCs (*p* < 0.001), Hb (*p* < 0.001), sCr (*p* < 0.001), and sodium (*p* < 0.001), and significantly higher prothrombin time (*p* < 0.001), total bilirubin (*p* = 0.001), aspartate aminotransferase (*p* < 0.001), alanine aminotransferase (*p* = 0.049), and RDW (*p* < 0.001) levels. There was no significant difference in platelet count (*p* = 0.163) and eGFR (*p* = 0.125) between the two groups.

The surgery time was significantly longer in the PNI <44 group (*p* = 0.028), and colloids were administered in a higher amount in the PNI <44 group (*p* = 0.002). There was no significant difference in the incidence of laparoscopic surgery (*p* = 0.392), total fluids (*p* = 0.729), and urine output (*p* = 0.469) between the two groups.

### 3.1. Primary Outcomes

The incidence of intraoperative transfusions according to the PNI (<44 and ≥44) is shown in Table 1. Out of the 1065 patients, 63 (5.9%) received intraoperative blood transfusions, 24 (2.3%) received massive transfusions, and 33 (3.1%) received postoperative transfusions. The intraoperative transfusions were 12.6% (42/333) in the PNI <44 group and 2.9% (21/732) in the PNI ≥44 group, with the differences between the two groups being significant (*p* < 0.001). The massive transfusion rate was 5.1% (17/333) in the PNI <44 group and 1.0% (7/732) in the PNI ≥44 group, differing significantly between the two groups (*p* < 0.001). There was no significant difference in postoperative transfusions between the two groups (*p* = 0.945).

In the multivariable analysis, preoperative PNI was significantly associated with intraoperative transfusion (odds ratio (OR) 2.54, 95%CI 1.21–5.32, *p* = 0.014) (Table 2). Additionally, the age (OR 1.05, 95%CI 1.01–1.08, *p* = 0.010), Hb level (OR 0.60, 95%CI 0.48–0.75, *p* < 0.001), operation time (OR 1.02, 95%CI 1.01–1.02, *p* < 0.001), synthetic colloid use (OR 2.34, 95%CI 1.13–4.82, *p* = 0.021), and laparoscopic surgery (OR 0.15, 95%CI 0.04–0.15, *p* = 0.003) were significantly associated with intraoperative transfusion (Table 2). However, preoperative PNI was not significantly associated with postoperative trans-fusion (odds ratio (OR) 1.92, 95%CI 0.90–4.10, *p* = 0.089) (Appendix A).

### 3.2. Secondary Outcomes

The PNI < 44 group demonstrated a higher incidence of PHLF (18.32% vs. 4.64%, *p* < 0.001); however, no significant differences were observed in hospital days (21.18 days vs. 20.79 days, *p* = 0.851), rates of postoperative AKI (7.21% vs. 5.60%, *p* = 0.334), ICU admissions (7.51% vs. 7.24%, *p* = 0.899), and prolonged ICU stay (4.20% vs. 2.46%, *p* = 0.125) (Table 1). The PNI <44 group also showed poor 1-year survival (12.01% vs. 3.42%, *p* < 0.001), 5-year survival (34.83% vs. 17.90%, *p* < 0.001), and overall survival (42.64% vs. 24.45%, *p* < 0.001) (Table 1). In the Cox regression analysis, preoperative PNI <44 was significantly associated with 5-year survival (hazards ratio [HR] 1.69, 95%CI 1.27–2.24, *p* < 0.001) (Table 3). MELD scores (HR 1.17, 95%CI 1.06–1.29, *p* = 0.002), TNM staging (stage 2; HR 1.57, 95%CI 0.90–2.76, *p* = 0.013, stage 3; HR 2.61, 95%CI 1.26–5.39, *p* = 0.010, stage 4; HR 3.72, 95%CI 2.84–4.88, *p* < 0.001), synthetic colloid use (HR 1.70, 95%CI 1.31–2.22, *p* < 0.001), and transfusion (HR 2.21, 95%CI 1.51–3.23, *p* < 0.001) were also significantly associated with the 5-year survival (Table 3).

Preoperative PNI < 44 was significantly associated with PHLF (OR 3.02, 95%CI 1.87–4.87, *p* = 0.002) and 1-year survival (HR 1.61, 95%CI 1.19–2.18, *p* = 0.002) even after adjusting for other potentially confounding variables (Table 4).

The addition of PNI to the clinical model for intraoperative transfusion consisting of age, MELD, and Hb level showed no significant improvement in AUC (*p* = 0.506) but showed significant improvement of the predictive power in NRI analysis (0.392, 95% CI 0.143–0.641, *p* = 0.002) (Table 5). The addition of PNI to the clinical model for 5-year survival consisting of age, sex, DM, TNM staging, MELD, synthetic colloid use, and RBC transfusion showed a significant improvement in the AUC (*p* = 0.021), which also showed significant improvement of the predictive power in NRI analysis (0.136, 95% CI 0.041–0.213, *p* = 0.004) (Table 5).

Figure 2 shows the Kaplan–Meier curve according to preoperative PNI (<44 and ≥44). The 5-year survival was significantly lower in the PNI level <44 group than in the PNI level ≥ 44 group (log-rank test; *p* < 0.001).

## 4. Discussion

This study demonstrated a significant difference in the incidence of intraoperative transfusion (12.61% vs. 2.87%) according to the PNI (<44 and ≥44) in patients who underwent hepatectomy for HCC. PNI <44 was also associated with PHLF and a poor survival rate. NRI analysis showed significant improvement in the predictive power of PNI. These main findings suggest that preoperative PNI level may be an independent predictor for intraoperative transfusion and surgical outcomes in HCC patients.

Intraoperative bleeding is a major surgical problem, resulting in increased morbidity and mortality [17,18]. Perioperative uncontrolled bleeding leads to a combination of hemodilution, hemostatic factor consumption, hypothermia, and acid base imbalance, which resulted in acquired coagulation abnormalities and a vicious cycle [19]. In addition, intraoperative blood transfusion is associated with a higher risk of mortality and morbidity in surgical patients [20]. Major surgery for liver diseases, such as partial hepatectomy and liver transplantation (LT), is associated with significant intraoperative bleeding and transfusion [21,22]; therefore, many surgical and anesthetic techniques have been used in an attempt to minimize bleeding during liver surgery [23,24]. However, intraoperative bleeding and transfusion during liver surgery still remains difficult to predict [25,26]. One study reported that the MELD score did not predict intraoperative bleeding and transfusion during liver transplantation [27]. There is a poor association between bleeding and prothrombin time in patients with chronic liver disease [28,29]. Although there have been studies showing that preoperative Hb is an important factor in predicting bleeding and transfusion during liver surgery [26,30,31], few studies have demonstrated the predictive power of other biologic markers for intraoperative transfusion in patients with HCC. Recently, one study reported that the PNI < 45.6 was significant associated with postoperative complications and blood loss in patients who underwent partial hepatectomy [32]. Our study is clinically meaningful as a major investigation of the association between preoperative PNI and intraoperative transfusion in patients undergoing hepatectomy for HCC. In this study, the incidence of intraoperative transfusion increased significantly in patients with a PNI cut-off value < 44. This is consistent with other previous studies suggesting 44 as the cut-off value for complications [32,33]. Additionally, the clinical model, combined with PNI and known risk factors, improved the predictive power for intraoperative blood transfusion. These results suggest that PNI is strongly associated with intraoperative transfusion.

In our multivariate logistic regression analysis, preoperative PNI < 44, age, operation time, laparoscopic surgery, synthetic colloid use, and Hb were significantly associated with intraoperative transfusion. The age and operation time reflect the patient’s systemic condition, comorbid disease status, and severity of disease, and have been reported as risk factors for blood transfusion in various surgeries [34,35,36]. Laparoscopic liver resection has been reported to have less blood loss and less need for transfusion with regard to the benefits of the minimally invasive approach [37]. Synthetic colloid administration has been reported to be associated with coagulopathy and transfusion [38,39]. However, in our study, the synthetic colloids were limited to 20 mL/kg; therefore, the relationship between synthetic colloids and transfusion is thought to be a consequence of the increased colloid use due to bleeding.

Preoperative Hb has been reported as a risk factor of intraoperative bleeding in liver surgery [26,30,31]. Transfusions occur when Hb levels are low; therefore, Hb is one of the strongest indicators of the need for transfusions. Previous studies have shown that there is no significant difference in oxygen delivery capacity, mortality, and morbidity when Hb is 7 or higher [40,41]. Therefore, restrictive transfusion approaches are increasingly being implemented as best practices in recent years [42].

In the current study, the association between preoperative PNI and transfusions seems to be due to characteristics of PNI, which reflect the patient’s inflammatory response and nutritional status. Various inflammatory reactions and release of cytokines interfere with the coagulation system and create a thrombotic state [43], which increases the risk of bleeding during surgery [44,45]. Nutritional deficiency can accelerate catabolic response, resulting in decreased protein synthesis and liver regeneration [46]. Deterioration of liver function and cirrhosis on its own makes surgery difficult and may increase the risk of bleeding, associated with a thrombotic state [47,48]. In addition, recent studies reported that elevated RDW may be associated with bleeding risk [49]. The suppression of erythrocyte maturation by inflammatory cytokines causes anisocytosis and abnormal erythropoietin function, which is associated with anemia, thrombotic state [50], and hemorrhagic tendency [49,51]. In this study, there was a significant difference in RDW values according to the PNI cut-off value; this may have a significant correlation with the findings of the study. However, in our multivariate logistic regression analysis, PNI was significantly associated with intraoperative transfusion, whereas RDW did not. Therefore, in our study, PNI seems to have a better association with blood transfusions than RDW.

In our study, patients with a PNI level < 44 had a higher incidence of PHLF and poor 1-year and 5-year survival rates. In addition, PNI showed better predictive power for survival when combined with other risk factors. These results suggest that low PNI levels provide better information in predicting poor surgical prognosis.

There are several limitations to our study. First, our study is a retrospective study, and confounding factors not considered may cause potential errors. However, we tried to reduce the impact of confounding factors by adjusting for variables that could affect the outcome. Second, our data were collected from a single medical center and the results may have been biased due to homogeneous groups. Therefore, further studies of heterogeneous groups are required. In addition, our center has achieved over 800 liver resections per year and 300 LDLTs per year since 2010 [52], and the liver transplant team has extensive experience; therefore, our transfusion rate may be rather low than in other centers. Third, in our study, the proportion of the high TNM stage (≥3) group is high (30.0%), which may affect the increase of mortality. However, nevertheless, the 1-year and 5-year mortality in our center are 6.1% and 23.2%, respectively, which are generally lower than those of other medical centers reported in the literature [53]. Fourth, to date, there is no exact consensus on the cutoff value of preoperative PNI that predicts intraoperative transfusion and survival rates. Therefore, more well-designed studies are needed to accurately validate preoperative PNI cutoff values.

## 5. Conclusions

In conclusion, a preoperative PNI level of <44 was associated with various surgical outcomes and intraoperative transfusions in patients who underwent hepatectomy. These results suggest that preoperative PNI might be a potent predictor of transfusion and surgical prognosis in HCC surgery.

## Figures and Tables

**Figure 1 cancers-13-02508-f001:**
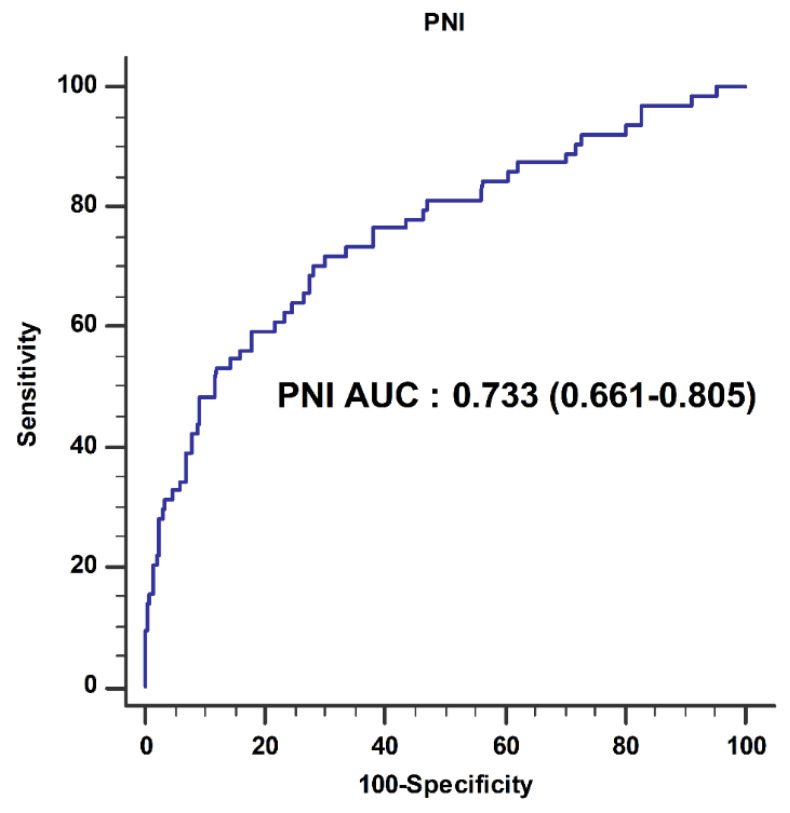
Receiver operating characteristic (ROC) curve analysis of the prognostic nutritional index (PNI) for predicting intraoperative transfusion.

**Figure 2 cancers-13-02508-f002:**
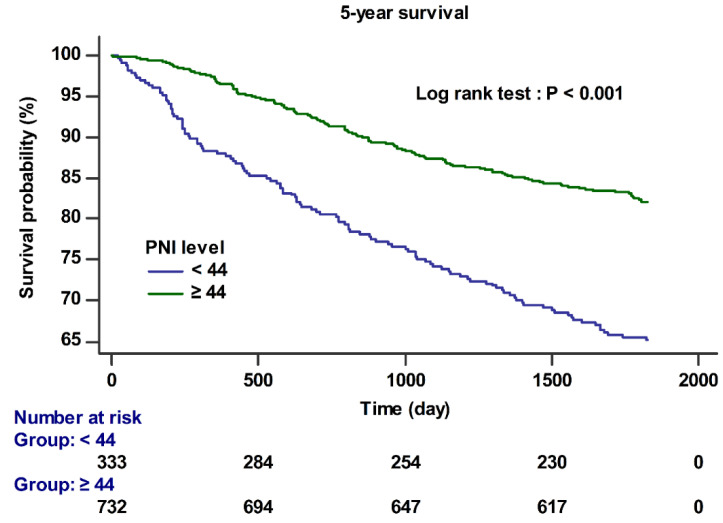
Kaplan–Meier survival curves for 5-year survival according to the preoperative PNI cut off value <44 (log-rank test; *p* < 0.001).

**Table 1 cancers-13-02508-t001:** Demographic, laboratory, and intraoperative variables, and surgical outcomes of the study population.

Demographic, Laboratory, and Intraoperative Variables			
	Prognostic Nutritional Index		
	<44 (*n* = 333)	≥44 (*n* = 732)	*p*-Value
**Demographic variables**			
Age; years	57.14 ± 10.67	55.06 ± 10.10	0.026
Sex; female	85 (25.53)	124 (16.94)	0.002
Weight; kg	64.21 ± 10.16	67.56 ± 10.17	0.185
BMI; kg.m^−2^	23.83 ± 3.00	24.33 ± 2.82	0.038
Liver cirrhosis	124 (37.24)	264 (36.7)	0.764
Etiology			0.623
HBV	258 (77.48)	540 (73.77)	
HCV	26 (7.81)	70 (9.56)	
Alcoholic	30 (9.01)	76 (10.38)	
NAFLD	19 (5.71)	46 (6.28)	
Antiviral therapy	143 (42.94)	320 (43.72)	0.866
TACE	66 (19.82)	132 (18.03)	0.542
TNM staging			0.046
1	194 (58.38)	491 (67.03)	
2	20 (5.99)	41 (5.61)	
3A	8 (2.4)	8 (1.09)	
3B	2 (0.6)	3 (0.41)	
4A	95 (28.44)	174 (23.8)	
4B	14 (4.19)	15 (2.05)	
Number of tumors			
Solitary	290 (87.09)	658 (89.89)	0.211
≥2	43 (12.91)	74 (10.11)	
Tumor size	5.61 ± 4.83	4.00 ± 2.97	<0.001
Lymph node invasion	103 (30.93)	183 (0.25)	0.025
Metastasis	14 (4.20)	16 (2.19)	0.100
DM	24 (7.21)	43 (5.87)	0.487
HTN	29 (8.71)	45 (6.15)	0.163
CAD	2 (0.60)	7 (0.96)	0.728
MELD scores	7.70 ± 1.42	7.01 ± 1.00	<0.001
CTP scores	5.72 ± 0.57	5.08 ± 0.27	<0.001
**Laboratory Variables**			
WBC	4.65 ± 1.84	5.74 ± 1.65	<0.001
Hemoglobin	12.98 ± 1.66	14.33 ± 1.40	<0.001
Platelets	152.20 ± 85.86	168.57 ± 55.21	0.163
Prothrombin time	1.07 ± 0.09	1.02 ± 0.06	<0.001
Creatinine; mg.dL^−1^	0.79 ± 0.18	0.84 ± 0.17	<0.001
eGFR; mL/min/1.73m^2^	68.70 ± 14.66	74.28 ± 13.09	0.125
Total bilirubin	0.84 ± 0.47	0.77 ± 0.34	0.001
AST	49.17 ± 45.70	35.66 ± 20.68	<0.001
ALT	37.38 ± 32.29	36.39 ± 25.55	0.049
Sodium	139.27 ± 2.96	140.10 ± 2.30	<0.001
RDW	13.52 ± 1.55	12.91 ± 1.03	<0.001
PNI	40.38 ± 3.18	49.77 ± 4.01	<0.001
**Intraoperative Variables**			
Operation time; min	273.83 ± 85.49	266.20 ± 76.29	0.028
Laparoscopic surgery	67 (20.12)	166 (22.68)	0.392
Extensive surgery (≥3 segments)	13 (3.90)	29 (3.96)	0.901
Total fluids; mL/kg	42.01 ± 22.02	38.06 ± 18.06	0.729
Colloid (mL/kg)	4.80 ± 7.05	4.61 ± 5.20	0.002
Synthetic colloid use	142 (42.64)	388 (53.01)	0.002
Urine output; mL/kg/hr	1.89 ± 1.38	1.67 ± 1.05	0.469
**Surgical outcomes of the study population**			
	**Prognostic Nutritional Index**		
	**<44 (*n* = 333)**	**≥44 (*n* = 732)**	***p*** **-Value**
**Transfusions**			
RBC transfusion	42 (12.61)	21 (2.87)	<0.001
FFP transfusion	5 (1.50)	2 (0.27)	0.034
Platelet transfusion	3 (0.90)	0 (0.00)	0.030
Massive transfusion (≥4 units)	17 (5.11)	7 (0.96)	<0.001
Postoperative transfusion	11 (3.30)	22 (3.00)	0.945
**Surgical outcomes**			
Hospital days	21.18 ± 14.14	20.79 ± 12.65	0.851
AKI	24 (7.21)	41 (5.60)	0.334
PHLF	61 (18.32)	34 (4.64)	<0.001
ICU admission	25 (7.51)	53 (7.24)	0.899
ICU stay (≥2 days)	14 (4.20)	18 (2.46)	0.125

BMI, body mass index; DM, diabetes mellitus; HTN, hypertension; CAD, coronary artery disease; MELD, model for end-stage liver disease; CTP, Child–Turcotte–Pugh; WBC, white blood cell; eGFR, estimated glomerular filter ratio; AST, aspartate aminotransferase; ALT, alanine aminotransferase; RDW, red cell distribution width; PNI, prognostic nutritional index; SD, standard deviation; RBC, red blood cell; AKI, acute kidney injury; PHLF, post hepatectomy liver failure; ICU, intensive care unit. Values are expressed as the mean ± SD, median (interquartile range), or n (proportion).

**Table 2 cancers-13-02508-t002:** Univariate and multivariable analysis of the risk factors of intraoperative transfusion.

	Univariate			Multivariate		
	OR	95% CI	*p*-Value	OR	95% CI	*p*-Value
PNI (<44)	5.25	3.04–9.08	<0.001	2.20	1.06–4.60	0.035
Age	1.04	1.02–1.07	0.002	1.04	1.01–1.08	0.010
Sex (male)	0.94	0.50–1.76	0.835			
BMI	0.87	0.79–0.96	0.004			
DM	1.62	0.67–3.92	0.280			
HTN	1.74	0.77–3.98	0.186			
MELD scores	1.37	1.15–1.63	<0.001	1.26	1.00–1.61	0.054
CTP scores	12.40	4.55–33.80	<0.001			
TNM staging			0.008			
1	1.00(Ref.)					
2	2.10	0.78–5.64	0.143			
3	1.17	0.15–9.06	0.878			
4	2.53	1.48–4.33	0.001			
Hemoglobin	0.57	0.49–0.67	<0.001	0.59	0.47–0.74	<0.001
RDW	1.34	1.16–1.53	<0.001	1.07	0.83–1.38	0.610
Operation time; min	1.01	1.01–1.02	<0.001	1.02	1.01–1.02	<0.001
Synthetic colloid use	3.45	1.91–6.25	<0.001	2.30	1.12–4.73	0.024
Extensive surgery (≥3 segments)	6.63	3.15–13.92	<0.001	1.36	0.48–3.85	0.561
Laparoscopic surgery	0.17	0.05–0.54	0.003	0.16	0.04–0.57	0.005

OR, odds ratio; CI, confidence interval; BMI, body mass index; DM, diabetes mellitus; HTN, hypertension; MELD, model for end-stage liver disease; CTP, Child–Turcotte–Pugh; RDW, red cell distribution width; PNI, prognostic nutritional index; SD, standard deviation. Values are expressed as the mean ± SD, median (interquartile range), or n (proportion).

**Table 3 cancers-13-02508-t003:** Cox regression analysis of risk factors of 5-year survival.

	Univariate			Multivariate		
	HR	95% CI	*p*-Value	HR	95% CI	*p*-Value
PNI (<44)	2.20	1.71–2.83	<0.001	1.68	1.27–2.24	<0.001
Age	1.00	0.99–1.01	0.642	0.99	0.98–1.01	0.301
Sex (male)	1.25	0.90–1.75	0.186	1.14	0.81–1.61	0.461
BMI	0.94	0.90–0.98	0.041			
DM	1.53	0.99–2.37	0.057	1.49	0.94–2.35	0.087
HTN	0.83	0.49–1.40	0.488			
CAD	0.94	0.23–3.75	0.924			
MELD scores	1.24	1.14–1.35	<0.001	1.17	1.06–1.29	0.002
CTP scores	2.66	1.32–5.39	0.006			
TNM staging			<0.001			<0.001
1	1.00(Ref.)			1.00(Ref.)		
2	1.65	0.94–2.88	0.081	1.57	0.90–2.76	0.013
3	3.06	1.49–6.29	0.002	2.61	1.26–5.39	0.010
4	3.72	2.86–4.85	<0.001	3.72	2.84–4.88	<0.001
Hemoglobin	0.85	0.79–0.92	<0.001			
RDW	1.16	1.09–1.22	<0.001			
Operation time; min	1.00	1.00–1.00	<0.001			
Synthetic Colloid use	1.49	1.16–1.92	0.002	1.70	1.31–2.22	<0.001
Laparoscopic surgery	0.50	0.35–0.72	<0.001			
Transfusion	4.00	2.83–5.66	<0.001	2.21	1.51–3.23	<0.001

HR, hazards ratio; CI, confidence interval; BMI, body mass index; DM, diabetes mellitus; HTN, hypertension; MELD, model for end-stage liver disease; CTP, Child–Turcotte–Pugh; RDW, red cell distribution width; PNI, prognostic nutritional index; SD, standard deviation. Values are expressed as the mean ± SD, median (interquartile range), or n (proportion).

**Table 4 cancers-13-02508-t004:** Transfusion and surgical outcomes adjusted by preoperative PNI level (<44).

	Univariate			Multivariate		
	**OR**	**95% CI**	***p*-Value**	**OR**	**95% CI ***	***p*-Value**
RBC transfusion	5.25	3.04–9.08	<0.001	2.20	1.06–4.60	0.035
PHLF	4.60	2.96–7.16	<0.001	3.02	1.87–4.87	<0.001
	**HR**	**95% CI**	***p*-value**	**HR**	**95% CI †**	***p*-value**
1-year survival	3.86	2.30–6.48	<0.001	2.98	1.66–5.38	<0.001
5-year survival	2.45	1.83–3.29	<0.001	1.68	1.27–2.24	<0.001

* Adjusted for age, MELD, hemoglobin, RDW, operation time, laparoscopic surgery, synthetic colloid use, and extensive surgery. † Adjusted for age, sex, DM, TNM staging, MELD, synthetic colloid use, and RBC transfusion. OR, odds ratio; HR, hazards ratio; CI, confidence interval; PNI, prognostic nutritional index; RBC, red blood cell; PHLF, post hepatectomy liver failure; MELD, model for end-stage liver disease; RDW, red cell distribution width; SD, standard deviation. Values are expressed as the mean ± SD, median (interquartile range), or n (proportion).

**Table 5 cancers-13-02508-t005:** Improvement in the AUC and NRI by adding PNI to the clinical predictive models.

		AUC (95% CI)	*p*-Value	NRI (95% CI)	*p*-Value
Transfusion	Model 1 *	0.757 (0.694–0.820)			
	Model 1 * + PNI	0.768 (0.702–0.834)	0.506	0.392 (0.143–0.641)	0.002
5-year survival	Model 2 †	0.725 (0.692–0.758)			
	Model 2 † + PNI	0.742 (0.710–0.773)	0.021	0.136 (0.041–0.213)	0.004

* Model 1 = age + MELD + hemoglobin. † Model 2 = age + sex + DM + TNM staging + MELD + synthetic colloid use + RBC transfusion. AUC, areas under the curves; CI, confidence interval; NRI, net reclassification index; PNI, prognostic nutritional index; MELD, model for end-stage liver disease; DM, diabetes mellitus; RBC, red blood cell; SD, standard deviation. Values are expressed as the mean ± SD, median (interquartile range), or n (proportion).

## Data Availability

The data presented in this study are available on request from the corresponding author. The data are not publicly available due to conditions of the ethics committee of our university.

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
