# Peer review of "The Association between Prognostic Nutritional Index (PNI) and Intraoperative Transfusion in Patients Undergoing Hepatectomy for Hepatocellular Carcinoma: A Retrospective Cohort Study"

_cancers, 2021, doi:10.3390/cancers13112508_

Round 1
Reviewer 1 Report
The authors introduced an evaluating prospect of the prognostic nutritional index (PNI) for patients with HCC who underwent hepatectomy, which is associated with significant intraoperative bleeding and transfusion. In a retrospective cohort study, the association between PNI and intraoperative transfusions was evaluated by different statistical methods. They confirmed the importance of already exposed PNI prognostic predictor in various cancers also in the case of HCC.
Minor comments:
In lines 43-45 the authors quoted otherwise an important etiological factor hepatitis C virus (HCV), but certainly not the only one, which has great importance for an increasing incidence of HCC in humans. In more than 80% of cases, HCC develops with cirrhosis-related etiological factors of viral hepatitis and non-alcoholic liver disease (NAFLD). While the incidence of HBV/HCV-related HCC is decreasing, a clear trend of increase in NAFLD-related HCC cases observed, especially in countries with a high socio-demographic index. I suggest reformulating the statement and add NAFLD as an additional important factor of influence on HCC development.
In line 76, please correct the hyphen in stating the concentration of rocutonium.
Line 147: the authors explained the p-values less than 0.05 were considered statistically significant. Were those p-values adjusted?
Line 161: The prospective PNI <44 group is considered more likely to be female (0.002), but in Table 1 I found written the male sex with the same p-value. Is this a mistake? The authors should check it.
Line 182: Should be ˝between˝ but not bet6ween˝.
Table 1: Could be improved by dividing into at least two-part (Table 1A and 1B): as it spreads between the two sides, it is harder to follow. Besides, in group <44 for ALT space is missing in values (37.38 ± 32.29).
Please unify the words PNI <44 and cutoff. In some places in the manuscript, the PNI <44 or cut-off are stated, respectively.
Table 5 should not be divided into two pages.
Author Response
Responses to Reviewer #1
- In lines 43-45 the authors quoted otherwise an important etiological factor hepatitis C virus (HCV), but certainly not the only one, which has great importance for an increasing incidence of HCC in humans. In more than 80% of cases, HCC develops with cirrhosis-related etiological factors of viral hepatitis and non-alcoholic liver disease (NAFLD). While the incidence of HBV/HCV-related HCC is decreasing, a clear trend of increase in NAFLD-related HCC cases observed, especially in countries with a high socio-demographic index. I suggest reformulating the statement and add NAFLD as an additional important factor of influence on HCC development.
Response: Thank you for the thoughtful suggestion. As per your recommendation, we have revised the previous sentence as follows: “The incidence of HCC is increasing due to hepatitis C virus (HCV) infection, with more than 782,000 new cases diagnosed each year, and recently, non-alcoholic fatty liver disease (NAFLD) has become an important cause of hepatocellular carcinoma in developed areas [1,2].” (page 2, lines 46–48).
- In line 76, please correct the hyphen in stating the concentration of rocuronium.
Response: Thank you for pointing out our error. We have revised the sentence as follows: “rocuronium (0.6–1.2 mg/kg).” (page 2, line 78).
- Line 147: the authors explained the p-values less than 0.05 were considered statistically significant. Were those p-values adjusted?
Response: Thank you for the question. We apologize for the incorrect description. The Kaplan–Meier (KM) method was used to analyze the cumulative survival rate between two groups (PNI <44 and PNI ≥44). Therefore, adjusted p-values were not required. We have revised the sentence as follows: “The Kaplan–Meier (KM) method was used to analyze the cumulative survival rate between two groups (PNI level <44 group and ≥44 group)” (page 4, lines 152–154).
- Line 161: The prospective PNI <44 group is considered more likely to be female (0.002), but in Table 1 I found written the male sex with the same p-value. Is this a mistake? The authors should check it.
Response: Thank you for asking. In our study, the proportion of women with PNI <44 is 25.5%, and the proportion of women with PNI ≥44 is 16.9%; therefore, the proportion of women is higher in the PNI <44 group (p=0.002). However, since Table 1 was somewhat confusing, we have revised it so that it refers to the female sex only.
- Line 182: Should be ˝between˝ but not bet6ween˝.
Response: Thanks for pointing out the error. We have corrected the typo (page 5, line 197).
- Table 1: Could be improved by dividing into at least two-part (Table 1A and 1B): as it spreads between the two sides, it is harder to follow. Besides, in group <44 for ALT space is missing in values (37.38 ± 32.29).
Response: Thank you for the thoughtful suggestion. As per your recommendation, we have divided Table 1 into Table 1A and Table 1B as shown below. Moreover, in group <44 for ALT, we have added the missing space as follows: “(37.38 ± 32.29).”
Table 1A. Demographic, laboratory, and intraoperative variables of the study population.
|
|
Prognostic nutritional index |
||
|
|
< 44 (n = 333) |
≥ 44 (n = 732) |
p-value |
|
Demographic variables |
|
|
|
|
Age; years |
57.14 ± 10.67 |
55.06 ± 10.10 |
0.026 |
|
Sex; female |
85 (25.53) |
124 (16.94) |
0.002 |
|
Weight; kg |
64.21 ± 10.16 |
67.56 ± 10.17 |
0.185 |
|
BMI; kg.m-2 |
23.83 ± 3.00 |
24.33 ± 2.82 |
0.038 |
|
Liver cirrhosis |
124 (37.24) |
264 (36.7) |
0.764 |
|
Etiology |
|
|
0.623 |
|
HBV |
258 (77.48) |
540 (73.77) |
|
|
HCV |
26 (7.81) |
70 (9.56) |
|
|
Alcoholic |
30 (9.01) |
76 (10.38) |
|
|
NAFLD |
19 (5.71) |
46 (6.28) |
|
|
Antiviral therapy |
143 (42.94) |
320 (43.72) |
0.866 |
|
TACE |
66 (19.82) |
132 (18.03) |
0.542 |
|
TNM staging |
|
|
0.046 |
|
1 |
194 (58.38) |
491 (67.03) |
|
|
2 |
20 (5.99) |
41 (5.61) |
|
|
3A |
8 (2.4) |
8 (1.09) |
|
|
3B |
2 (0.6) |
3 (0.41) |
|
|
4A |
95 (28.44) |
174 (23.8) |
|
|
4B |
14 (4.19) |
15 (2.05) |
|
|
Number of tumors |
|
|
|
|
Solitary |
290 (87.09) |
658 (89.89) |
0.211 |
|
≥ 2 |
43 (12.91) |
74 (10.11) |
|
|
Tumor size |
5.61 ± 4.83 |
4.00 ± 2.97 |
< 0.001 |
|
Lymph node invasion |
103 (30.93) |
183 (0.25) |
0.025 |
|
Metastasis |
14 (4.20) |
16 (2.19) |
0.100 |
|
DM |
24 (7.21) |
43 (5.87) |
0.487 |
|
HTN |
29 (8.71) |
45 (6.15) |
0.163 |
|
CAD |
2 (0.60) |
7 (0.96) |
0.728 |
|
MELD scores |
7.70 ± 1.42 |
7.01 ± 1.00 |
< 0.001 |
|
CTP scores |
5.72 ± 0.57 |
5.08 ± 0.27 |
< 0.001 |
|
Laboratory Variables |
|
|
|
|
WBC |
4.65 ± 1.84 |
5.74 ± 1.65 |
< 0.001 |
|
Hemoglobin |
12.98 ± 1.66 |
14.33 ± 1.40 |
< 0.001 |
|
Platelets |
152.20 ± 85.86 |
168.57 ± 55.21 |
0.163 |
|
Prothrombin time |
1.07 ± 0.09 |
1.02 ± 0.06 |
< 0.001 |
|
Creatinine; mg.dl−1 |
0.79 ± 0.18 |
0.84 ± 0.17 |
< 0.001 |
|
eGFR; mL/min/1.73m2 |
68.70 ± 14.66 |
74.28 ± 13.09 |
0.125 |
|
Albumin; g.dl−1 |
3.40 ± 0.31 |
3.97 ± 0.31 |
< 0.001 |
|
Total bilirubin |
0.84 ± 0.47 |
0.77 ± 0.34 |
0.001 |
|
AST |
49.17 ± 45.70 |
35.66 ± 20.68 |
< 0.001 |
|
ALT |
37.38 ± 32.29 |
36.39 ± 25.55 |
0.049 |
|
Sodium |
139.27 ± 2.96 |
140.10 ± 2.30 |
< 0.001 |
|
RDW |
13.52 ± 1.55 |
12.91 ± 1.03 |
< 0.001 |
|
PNI |
40.38 ± 3.18 |
49.77 ± 4.01 |
< 0.001 |
|
Intraoperative Variables |
|
|
|
|
Operation time; min |
273.83 ± 85.49 |
266.20 ± 76.29 |
0.028 |
|
Laparoscopic surgery |
67 (20.12) |
166 (22.68) |
0.392 |
|
Extensive surgery |
13 (3.90) |
29 (3.96) |
0.901 |
|
Total fluids; ml/kg |
42.01 ± 22.02 |
38.06 ± 18.06 |
0.729 |
|
Colloid (ml/kg) |
4.80 ± 7.05 |
4.61 ± 5.20 |
0.002 |
|
Synthetic colloid use |
142 (42.64) |
388 (53.01) |
0.002 |
|
Urine output; ml/kg/hr |
1.89 ± 1.38 |
1.67 ± 1.05 |
0.469 |
Table 1B. Surgical outcomes of the study population.
|
|
Prognostic nutritional index |
||
|
|
< 44 (n = 333) |
≥ 44 (n = 732) |
p-value |
|
Transfusions |
|
|
|
|
RBC transfusion |
42 (12.61) |
21 (2.87) |
<0.001 |
|
FFP transfusion |
5 (1.50) |
2 (0.27) |
0.034 |
|
Platelet transfusion |
3 (0.90) |
0 (0.00) |
0.030 |
|
Massive transfusion (≥4 units) |
17 (5.11) |
7 (0.96) |
<0.001 |
|
Postoperative transfusion |
11 (3.30) |
22 (3.00) |
0.945 |
|
Surgical outcomes |
|
|
|
|
Hospital days |
21.18 ± 14.14 |
20.79 ± 12.65 |
0.851 |
|
AKI |
24 (7.21) |
41 (5.60) |
0.334 |
|
PHLF |
61 (18.32) |
34 (4.64) |
< 0.001 |
|
ICU admission |
25 (7.51) |
53 (7.24) |
0.899 |
|
ICU stay (≥ 2 days) |
14 (4.20) |
18 (2.46) |
0.125 |
BMI, body mass index; DM, diabetes mellitus; HTN, hypertension; CAD, coronary artery disease; MELD, model for end-stage liver disease; CTP, Child-Turcotte-Pugh; WBC, white blood cell; eGFR, estimated glomerular filter ratio; AST, aspartate aminotransferase; ALT, alanine aminotransferase; RDW, red cell distribution width; PNI, prognostic nutritional index; SD, standard deviation; RBC, red blood cell; AKI, acute kidney injury; PHLF, post-hepatectomy liver failure; ICU, intensive care unit. Values are expressed as the mean ± SD, median (interquartile range), or n (proportion).
- Please unify the words PNI <44 and cutoff. In some places in the manuscript, the PNI <44 or cut-off are stated, respectively.
Response: Thank you for the recommendation. We changed the word “cutoff” to “PNI <44” throughout the manuscript to avoid confusion.
- Table 5 should not be divided into two pages.
Response: Thank you for the thoughtful suggestion. As per your recommendation, we have presented Table 5 on one page in the text.

Reviewer 2 Report
Dear Authors,
the present manuscript presents a robust study , evaluating the relationship of the prognostic nutritional index (PNI) and intraoperative transfusions during hepatectomy. As some spelling error are evident in the manuscript I suggest a minor text editing prior to publication.
Author Response
Response to Reviewer #2
- the present manuscript presents a robust study, evaluating the relationship of the prognostic nutritional index (PNI) and intraoperative transfusions during hepatectomy. As some spelling errors are evident in the manuscript. I suggest a minor text editing prior to publication.
Response: We appreciate your comments. We have corrected the spelling errors in the previous manuscript.

Reviewer 3 Report
Please see the enclosed file

Author Response
Responses to Reviewer #3
- PNI is calculated from albumin and leucocytes count. Both parameters are expected to be lower in patients with cirrhosis, especially in the setting of portal hypertension. This is important information and should be added in Table 1.
Response: We agree with you. We have included the proportions of liver cirrhosis according to PNI <44 (37.2%, 124/333) and PNI ≥44 (36.1%, 264/732) in Table 1. There was no significant difference between the two groups (p=0.764).
- The epidemiology of HCC varies from a region of the world to another. While HCC is mainly related to viral hepatitis in Asia, NAFLD and alcohol abuse is a major risk factor in the western hemisphere. Viral hepatitis and antiviral therapy can have a major impact on the immune system and leucocytes count of the affected host. There is no precision on the etiology of cirrhosis/HCC and antiviral treatment in this study.
Response: Thank you for pointing out this issue. We fully agree with your opinion, and therefore, we have added data on the etiology of cirrhosis/HCC and antiviral therapy to Table 1A, since these parameters are major potential confounders and could have a significant impact on the findings. However, the etiology of cirrhosis and HCC was not significantly different between the two groups (PNI <44 and PNI ≥44).
- Most literature on the subject PNI and HCC looked at the disease-free survival and overall survival after hepatectomy. In this study, SIM and all looked at the impact of PNI on intraoperative need of transfusion. Coagulation in patients with liver disease is complex and still not fully understood. PNI could be an interesting tool to help estimate bleeding risk in patients with liver disease. PNI was calculated using AUC. A PNI of 44 was properly selected as a cut-off. This is similar to other studies on the subject, lending validity to this cohort. PNI < 44 was strongly associated with the need of intraoperative transfusion both in the univariate and multivariate analysis when assessing for the MELD score, age, HB level, operation time, synthetic colloid use and laparoscopic surgery. However, this endpoint is of limited clinical relevance.
- Post-operative bleeding is major cause of morbidity and mortality after hepatectomy. Looking at post-operative transfusion could have given us more information on the utility of PNI when assessing a patient bleeding risk.
- We have no information on the use of other blood products for coagulation such as fresh frozen plasma, platelets, etc
- Extended hepatectomy (more than 3 segments) and tumor vascular invasion are also classic risk factors associated with intraoperative need of transfusion and was not assessed in this study. Therefore, the extent of hepatectomy needs to be taken into account before this paper is being accepted.
Response: Thank you for the insightful comments.
- We have included data on postoperative transfusion in Table 1. We have also added the results of the univariate and multivariate analyses of the risk factors of postoperative transfusion to Supplementary Table 1.
Supplementary Table 1. Results of univariate and multivariate analyses of the risk factors of postoperative transfusion.
|
|
Univariate |
Multivariate |
||||
|
|
OR |
95% CI |
P–value |
OR |
95% CI |
P–value |
|
PNI (< 44) |
2.41 |
1.20–4.83 |
0.014 |
1.92 |
0.90–4.10 |
0.089 |
|
Age |
0.99 |
0.96–1.03 |
0.727 |
|
|
|
|
Sex (male) |
1.32 |
0.59–2.98 |
0.499 |
|
|
|
|
BMI |
0.93 |
0.82–1.05 |
0.242 |
|
|
|
|
DM |
0.45 |
0.06–3.40 |
0.445 |
|
|
|
|
HTN |
1.35 |
0.40–4.54 |
0.624 |
|
|
|
|
MELD scores |
1.37 |
1.09–1.71 |
0.007 |
1.27 |
0.98–1.64 |
0.066 |
|
CTP scores |
2.15 |
1.23–3.75 |
0.007 |
|
|
|
|
TNM staging |
|
|
0.070 |
|
|
|
|
1 |
1.00(Ref.) |
|
|
|
|
|
|
2 |
3.73 |
1.32–10.57 |
0.013 |
|
|
|
|
3 |
4.40 |
0.94–20.51 |
0.059 |
|
|
|
|
4 |
1.45 |
0.65–3.24 |
0.362 |
|
|
|
|
Hemoglobin |
0.84 |
0.68–1.03 |
0.100 |
|
|
|
|
RDW |
1.20 |
1.02–1.42 |
0.030 |
|
|
|
|
Operation time; min |
1.01 |
1.01–1.01 |
< 0.001 |
1.01 |
1.01–1.01 |
< 0.001 |
|
Synthetic colloid use |
2.38 |
1.12–5.05 |
0.024 |
1.97 |
0.90–4.30 |
0.090 |
|
Extensive surgery |
1.60 |
0.37–6.92 |
0.529 |
|
|
|
|
Laparoscopic surgery |
0.63 |
0.24–1.65 |
0.347 |
|
|
|
- We have added information on the use of FFP and platelets to Table 1B.
- We have assessed the information on extended hepatectomy (more than three segments) in Table 1A and included it in the results of the multivariate logistic analysis of intraoperative transfusion (Table 2). However, it did not significantly affect our previous results.
- The secondary outcomes of this study focus on the 1-year and 5-year survival of patients in function of the PNI. Authors need to be careful when drawing conclusion on data collected as a secondary issue, as their study was not designed for this. Overall, 1-year and 5-year survival was lower in patients with low PNI. This matches the available literature on the subject.
- Hepatectomy is generally reserved for patients with adequate liver functional reserve and early stage HCC (BCLC stage 0/A). In this study, 30% of patients had advanced stage HCC (stage 3 or 4). Advanced HCC is associated with more procedural complications, recurrence of disease and worst survival. Can the authors comment on this issue? What was the use of adjuvant therapy such a TACE before surgery?
- This study showed a high amount of primary liver failure and a low 5-year overall survival in this cohort. This could be secondary to the high tumor burden/cancer recurrence. This needs to be addressed in the discussion.
Response: Thank you for the insightful comments. In our study, approximately 30.0% (319/1065) of all patients belonged to the high TNM stage (≥3) group, 18.6% (198/1065) of all patients received preoperative TACE, and the 1-year and 5-year mortality of the high TNM stage were 15.7% (50/319) and 42.0% (134/319), respectively. As your comments indicate, mortality may have been affected by the proportion of high TNM stage patients. Nevertheless, 1-year and 5-year mortality in our center are 6.1% and 23.2%, respectively, which are generally lower than those of other medical centers reported in the literature.
- Lin CW, Chen YS, Lin CC, et al. Significant predictors of overall survival in patients with hepatocellular carcinoma after surgical resection. PLoS One. 2018;13:e0202650.
- Chok KSH, Chan MMY, Dai WC, et al. Survival outcomes of hepatocellular carcinoma resection with postoperative complications - a propensity-score-matched analysis. Medicine (Baltimore). 2017;96:e6430.
- Liu W, Wang K, Bao Q, Sun Y, Xing BC. Hepatic resection provided long-term survival for patients with intermediate and advanced-stage resectable hepatocellular carcinoma. World J Surg Oncol. 2016;14:62.
We have addressed this issue in the limitations section as follows: “Third, in our study, the proportion of the high TNM stage (≥3) group is high (30.0%, 319/1065), which may affect the mortality. However, 1-year and 5-year mortality in our center are 6.1% and 23.2%, respectively, which are generally lower than those of other medical centers reported in the literature.” (page 12, lines 345–349).
- Table 1: 1-year, 5-year and overall survival are presented as 1-year and 5-year mortality.
This is confusing. This data is already presented in the Kaplan-Meier curves and should be
removed from table 1.
Response: We have changed the word “survival” to “mortality” throughout the manuscript. Moreover, we have removed the data on mortality from Table 1.
- Some data presented in table 2 are redundant, this needs to be corrected.
Response: Thanks for the suggestion. The duplicated variable was our mistake, and we have deleted it.
- Table 4 is confusing. RBC transfusion and PHLF should be added to table 3. HR of survival
is already presented in the Kaplan-Meier curves.
Response: We apologize for the confusion. We have performed an additional logistic regression analysis and Cox regression analysis for PHLF and 1-year survival, respectively, and the results are summarized in Table 4. Therefore, Table 4 shows that preoperative PNI <44 was significantly associated with PHLF (OR 3.02, 95% CI 1.87–4.87, p<0.001) and 1-year survival (HR 2.98, 95% CI 1.66–5.38, p<0.001) even after adjusting for other potentially confounding variables. We have modified the title of Table 4 accordingly as follows: “Transfusion and surgical outcomes adjusted by preoperative PNI level (< 44).” (page 8, line 238).
- Line 208: The PNI < 44 group demonstrated a lower incidence of PHLF: it should read”
higher incidence”.
Response: Thank you for pointing out our error. We have corrected the text as follows: “The PNI <44 group demonstrated a higher incidence of PHLF (18.32% vs. 4.64%, p<0.001);” (page 8, lines 217–218).
- Typos are found in the short summary and need to be corrected.
Response: Thank you for the assessment. We have corrected the simple summary as follows: “The prognostic nutritional index (PNI), which describes a patient's nutritional status, inflammatory status, and immune response, has been reported as a predictor associated with prognosis in a variety of cancers and has been reported to be associated with surgical outcomes in patients with hepatocellular carcinoma (HCC). However, few studies have assessed the association between PNI and intraoperative transfusion. This study evaluated the predictive value of preoperative PNI for intraoperative transfusion in patients who had undergone hepatectomy for HCC. We found that preoperative PNI <44 was significantly associated with intraoperative transfusion and surgical outcomes. These results suggest that preoperative PNI might be a predictor of intraoperative transfusion and surgical prognosis in patients who have undergone hepatectomy.” (page 1, lines 13–21).
- Pringle maneuver is not a recent advance
Response: Thank you for your comment. We have improved the previous sentence as follows: “Although various surgical techniques, such as laparoscopy, the pringle maneuver, and inferior vena cava (IVC) clamping to minimize bleeding during liver surgery, have reduced intraoperative bleeding [3,4], hepatectomy still has a very high bleeding risk and often requires blood transfusion.” (page 2, lines 49–52).
- Please explain the reasons for excluding 102 patients from the analysis
Response: A total of 102 patients were excluded from the study (patients who had hematologic disease (N=4); patients who had taken anticoagulants, such as warfarin and aspirin (N=42); patients who had undergone emergency surgery (N=12); and patients with incomplete data or missing PNI values (N=44)). The exclusion criteria are mentioned in the Study design & patient population section, and we revised the previous sentence as follows: “Out of 1,167 patients who had undergone hepatectomy, 102 patients were excluded (patients who had hematologic disease [n=4]; patients who had taken anticoagulants, such as warfarin and aspirin [n=42]; patients who had undergone emergency surgery [n=12]; and patients with incomplete data or missing PNI values [n=44]).” (page 4, lines 159–162).
12.Since albumin is part of the PNI, it should not be analyzed. Furthermore, please do not
write that the PNI<44 has lower PNI…
Response: We appreciate the suggestion, and we have removed the description of albumin from Table 1 and Table 2. We have improved the text as follows: “Regarding the laboratory variables, the PNI <44 group had significantly lower levels of WBCs (p<0.001), Hb (p<0.001), sCr (p<0.001), and sodium (p<0.001) and significantly higher prothrombin time (p<0.001) as well as higher total bilirubin (p=0.001), aspartate aminotransferase (p<0.001), alanine aminotransferase (p=0.049), and RDW (p<0.001) levels.” (page 5, lines 178–181).
13.Comment on the frequency of patients who required transfusions: 6% seems low to me
Response: Thank you for your comment. Asan Medical Center in Korea has performed approximately 800 hepatic resections per year and more than 300 living donor liver transplantations per year since 2010. The transfusion rate during liver surgery in our center is reported to be about 2–10% according to the literature.
- Moon DB, Lee SG, Hwang S, et al. More than 300 consecutive living donor liver transplants a year at a single center. Transplant Proc. 2013;45:1942–1947.
- https://eng.amc.seoul.kr/gb/lang/specialities/illness.do?code=CA_Liver_Cancer
- Comparison of laparoscopic versus open left hemihepatectomy for left-sided hepatolithiasis
- Surgical outcomes following laparoscopic major hepatectomy for various liver diseases
- Factors associated with blood transfusion in donor hepatectomy: results from 2344 donors at a large single center
We have added the following sentences to the limitation section: “In addition, our center has performed over 800 liver resections per year and 300 LDLTs per year since 2010. As the liver transplant team has extensive experience, our transfusion rate may be lower than those in other centers.” (page 11, lines 343–345).
- Why did you decide to add DM to your model since you found no effect on your endpoints?
Response: Thank you for your question. In the Cox regression analysis, DM was not a significant variable in our study, but in several studies, DM has been reported as a factor associated with survival, so we added it to the final model.
- Chen S, Tao M, Zhao L, Zhang X. The association between diabetes/hyperglycemia and the prognosis of cervical cancer patients: A systematic review and meta-analysis. Medicine (Baltimore). 2017;96:e7981, doi:10.1097/md.0000000000007981.
- Gillani SW, Zaghloul HA, Ansari IA, et al. Multivariate analysis on the effects of diabetes and related clinical parameters on cervical cancer survival probability. Sci Rep. 2019;9:1084, doi:10.1038/s41598-018-37694-1.
- Define and comment on NRI: not sure that this is useful in your analysis. It sounds as if it
was coming from nowhere.
Response: The Net Reclassification Index (NRI) is a very popular measure for evaluating the improvement in prediction performance gained by adding a marker to a set of baseline predictors. NRI was used to comprehensively evaluate the discrimination of the model along with the AUC. Although in our study, there was no significant increase in AUC when PNI was added to the clinical transfusion model, NRI showed a significant increase in predictive power. In addition, when PNI was added to the clinical 5-year mortality model, both AUC and NRI showed a significant increase in predictive power. These results suggest the possibility of PNI as a predictor of intraoperative blood transfusion and survival in hepatectomy. We have added the definition of NRI and references to the Statistical analysis section as follows: “NRI is a measure for evaluating the improvement in prediction performance gained by adding a marker to a set of baseline predictors [16]. NRI was used to comprehensively evaluate the discrimination of the model along with the AUC [16].” (page 4, lines 145–148). We have also revised the previous sentence as follows: “The NRI analysis showed a significant improvement in the predictive power of PNI for transfusion (p=0.002) and 5-year survival (p=0.004)” (page 1, lines 34–35).